# Microbiological Profile of Fracture Related Infection at a UK Major Trauma Centre

**DOI:** 10.3390/antibiotics12091358

**Published:** 2023-08-24

**Authors:** Kavi H. Patel, Laura I. Gill, Elizabeth K. Tissingh, Athanasios Galanis, Ioannis Hadjihannas, Alexis D. Iliadis, Nima Heidari, Benny Cherian, Caryn Rosmarin, Alexandros Vris

**Affiliations:** 1Limb Reconstruction and Bone Infection Service, The Royal London Hospital, Barts Health NHS Trust, Whitechapel Road, London E1 1FR, UK; elizabeth.tissingh1@nhs.net (E.K.T.); alexis-dimitris.iliadis@nhs.net (A.D.I.); nima.heidari@nhs.net (N.H.); avris@nhs.net (A.V.); 2Department of Infection, The Royal London Hospital, Barts Health NHS Trust, Whitechapel Road, London E1 1FR, UK; laura.gill19@nhs.net (L.I.G.); benny.cherian@nhs.net (B.C.); crosmarin@nhs.net (C.R.); 3KAT General Hospital, Nikis 2, 14561 Kifisia, Greece; athanasios.galanis@nhs.net; 4Barts and the London School of Medicine, Garrod Building, Turner St., London E1 2AD, UK; i.hadjihannas@smd19.qmul.ac.uk

**Keywords:** fracture related infection, osteomyelitis, microbiology, anti-microbiological resistance, antibiotic guidelines

## Abstract

Fracture Related Infection (FRI) represents one of the biggest challenges for Trauma and Orthopaedic surgery. A better understanding of the microbiological profile should assist with decision-making and optimising outcomes. Our primary aim was to report on the microbiological profile of FRI cases treated over a six-year period at one of Europe’s busiest trauma centres. Secondarily, we sought to correlate our findings with existing anti-microbiological protocols and report on diagnostic techniques employed in our practice. All adult cases of FRI treated in our institution between 2016 and 2021 were identified, retrospectively. We recorded patient demographics, diagnostic strategies, causative organisms and antibiotic susceptibilities. There were 330 infection episodes in 294 patients. A total of 463 potentially pathogenic organisms (78 different species) were identified from cultures, of which 57.2% were gram-positive and 39.7% gram-negative. Polymicrobial cultures were found in 33.6% of cases and no causative organism was found in 17.5%. The most prevalent organisms were *Staphylococcus aureus* (24.4%), coagulase-negative Staphylococci (14%), *Pseudomonas aeruginosa* (8.2%), Enterobacter species (7.8%) and *Escherichia coli* (6.9%). Resistant gram-positive organisms (methicillin resistant *Staphylococcus aureus* or vancomycin-resistant Enterococci) were implicated in 3.3% of infection episodes and resistant gram-negatives (extended-spectrum beta-lactamase, ampC or carbapenemase-producing bacteria) in 13.6%. The organisms cultured in 96.3% of infection episodes would have been covered by our empirical systemic antibiotic choice of teicoplanin and meropenem. To our knowledge, this is the largest reported single-centre cohort of FRIs from a major trauma centre. Our results demonstrate patterns in microbiological profiles that should serve to inform the decision-making process regarding antibiotic choices for both prophylaxis and treatment.

## 1. Introduction

Fracture-related infection (FRI) is a severe complication following trauma and orthopaedic surgery and is becoming increasingly common [1]. It places a significant burden on healthcare systems with high direct and indirect costs [2]. Effective and timely treatment is crucial to minimize the devastating impact of this condition on patients. 

The treatment principles have been outlined in recent consensus guidelines and include accurate and timely diagnosis [1], surgical sampling [3], debridement (with stabilisation if required) [4,5] and targeted antimicrobial therapy [6]. An understanding of the microbial profile of FRI is crucial in providing optimal intra-focal and systemic antimicrobial therapy. This is especially relevant in those cases treated empirically or when culture results are negative. It is worth noting that in many settings the infrastructure needed to identify causative organisms is inadequate and this is a risk factor for recurrence [7].

Large multicentre studies focused on the microbiology of FRI are lacking. In addition, the term FRI is relatively new and the literature around FRI, osteomyelitis, open fractures and surgical site infection is often heterogenous in its definitions and descriptions. Current literature on the microbial profile of FRI shows wide variability but certain trends are encountered. *Staphylococcus aureus* (*S. aureus*) is the most common organism in FRI reported in the literature from both high and low-resource settings although this is a heterogeneous group of patients and pathologies [8,9,10,11].

The organisms involved in FRI following open fractures are a particular consideration. A case series of 310 Gustilo Anderson Grade III open fractures reported *S. aureus* as the most common organism with 28% of episodes growing an organism that was resistant to the prophylactic antibiotic agent prescribed on admission following injury [12]. The authors were unable to identify clinical risk factors predictive of causative organisms displaying resistance to their standard antibiotic regime.

In another study of 123 open fractures from Germany, the authors reported that most FRI cases were caused by coagulase-negative staphylococci (CoNS) and that the EAST (Eastern Association for the Surgery of Trauma) recommendations for antimicrobial cover in open fractures were adequate for all but one of these cases [13]. Reporting on a cohort of 213 combat injuries Burns et al. found a 27% infection rate with the most common causative organisms being gram-positive [14]. Both studies have looked at surveillance sampling of open fractures but surveillance culture results were shown to correlate poorly with infective culture results [13,14]. It is also worth highlighting that the organisms present at the initial time of contamination with an open fracture may be different from the organisms causing the FRI, which may be nosocomial [15]. 

There may be differences in antimicrobial profile between FRI and prosthetic joint infection (PJI). In a cohort of 167 patients with FRI and PJI, *S. aureus* was the most common organism in FRI but overall, there was no statistical difference in pathogen distribution between groups. Difficult-to-treat pathogens were more frequently detected in the PJI group (11.6 vs. 2.3%) [16]. 

Recent evidence has challenged the notion that the time of FRI presentation from initial injury can influence the microbial profile [17]. A study by Corrigan et al. found that early (<2 weeks) or delayed (3–10 weeks) FRI are most likely to be polymicrobial, whereas late (>10 weeks) FRIs are more likely to be culture-negative or monomicrobial. *S. aureus* was still the most common organism at all time points [17]. Similarly, Depypere et al. revealed that polymicrobial infections were more frequent in early FRI although they did not feel a time-based classification was useful in guiding antibiotic therapy or in evaluating microbial profile [18]. 

A better understanding of FRI microbial profiles encountered in different cohorts will allow optimal antibiotic selection both for prophylaxis (open fractures and intraoperative cover) as well as for empirical treatment in cases of FRI where culture results from intra-operative samples are pending. Recommendations exist at national and institutional levels, where possible, based on known common organisms and resistance profiles. In the UK, national guidelines exist for antimicrobial prophylaxis in open fractures although they allow for local modifications [19]. In our centre, the antibiotic choice in the initial stages of open fracture management is 1.2 g co-amoxiclav and empirical FRI treatment, pending culture results, is teicoplanin and meropenem. 

Antimicrobial resistance is a growing concern worldwide [20]. Resistant *Staphylococcus aureus* strains are of particular concern in low-resource settings. Vijayakumar et al. in a cohort of 100 patients with *Staphylococcus aureus*, report 75% of isolates were resistant to gentamicin, 81% to ciprofloxacin and 50% were methicillin-resistant *Staphylococcus aureus* (MRSA) [10]. In work describing two patient cohorts from 2001–2004 and 2013–2017 similar pathogens are described but they note the fall in MRSA from 11.4% to 8.3% of the *Staphylococcus aureus* isolates identified [21]. Rigorous antimicrobial stewardship will be essential to ensure that resistance patterns do not worsen and optimizing empiric guidelines should assist towards this.

The primary aim of this study was to describe the microbiological profile of FRI cases treated at a UK major trauma centre. The secondary aim was to present antimicrobial susceptibilities and comment on appropriate antimicrobial guidelines. We also report and comment on diagnostic and sampling strategies employed. 

## 2. Results

Over the study period, 294 patients with FRI were treated at our unit, of which 31 had more than one infection episode at least 30 days apart (recurrence at same anatomical location), contributing to a total of 330 infection episodes. Five patients were treated for FRI but did not have any surgical samples taken and were therefore removed from data analysis. Of the 325 episodes where surgical samples were sent, there were positive cultures in 268 (82.5%) and, overall, 78 distinct organisms were isolated. Patient demographics and co-morbidities are presented in Table 1. 

The anatomical distribution of cases is shown in Figure 1 with the tibia/fibula being the most common. We found no statistically significant difference in microbiology distribution with anatomical location, although one single case of *Cutibacterium acnes* was found in the shoulder region.

Our main finding was that *S. aureus* was the most isolated organism in our cohort of FRI patients, implicated in 34.2% of infection episodes; 7.1% of these were MRSA. Overall, gram-negative bacteria (GNB) accounted for 39.7% of microorganisms cultured, with the most common being *Pseudomonas aeruginosa* (8.2% of isolates) followed by *Enterobacter* species (7.8%) and *Escherichia coli* (6.9%). A breakdown of organisms identified (Table 2) with a graphical representation of the most common organisms (Figure 2) are also given. 

Our microbiological profile and resistance data used to assess predicted efficacy of our antibiotic therapy are presented in Table 3, Table 4 and Table 5. In our cohort, 16S PCR was used in 126 (38.8%) infection episodes with deep samples taken and swabs were performed in 96 episodes (29.1%). 

## 3. Discussion

Our primary aim was to describe the microbiological profile of FRI cases at a UK major trauma centre. The secondary aim was to review antimicrobial susceptibilities and comment on whether our guidelines for empirical systemic and intra-focal antibiotic therapy were effective in our cohort of FRI patients. 

The importance of *S. aureus* is reflected in other large studies (Sheehy et al. 32%, Ferreira et al. 39%) but these report on heterogenous groups with chronic osteomyelitis from all causes [22,23]. Waldvogel et al. reported *S. aureus* in approximately 60% of cases; however, the relative incidence of staphylococcal infections has fallen in subsequent studies perhaps due to a rise in contiguous infections [16,24]. A history of fracture or metalwork in-situ at the time of surgery is often associated with a broad range of organisms [25]. In this series, polymicrobial cultures were frequent (34.2% of cases) and higher than in previous studies from the developed world with Trampuz et al. reporting 27% (purely in patients with infections associated with fracture-fixation devices), Sheehy et al. 29% (although this study grouped both haematogenous and contiguous infections) [16,22] and Depypere et al. 25.3% [18].

A South African study showed a prevalence of 45% for polymicrobial infections and found that isolated gram-negative infections were most common in the post-traumatic group of patients [11]. Our overall incidence of GNB (39.7%) is significantly higher than other studies (Depypere et al. 26.2% and 27.8%) although all have found GNB to be the predominant pathogen in polymicrobial infection [18,26]. A potential explanation may be that, compared to similar studies, a higher proportion of our FRIs were related to a previous open fracture (37.4% vs. 24.7%) [18].

Methicillin-resistant *S. aureus* (MRSA), enteric gram-negative bacilli and Coagulase-negative staphylococci (CoNS) have been shown to be commonly associated with hospital-acquired infections [22]. Whilst CoNS are commensal pathogens, they have been shown to display high rates of resistance to antibiotics of clinical relevance and with their inherent ability to produce a potent biofilm, via polysaccharide intercellular adhesin, can be challenging to treat [27]. CoNS accounted for 14% of cultured organisms in our cohort.

Overall, our empirical antibiotic therapy of teicoplanin and meropenem was effective in 96.3% of episodes. Sixteen patients had infections involving gram-negative organisms resistant to gentamicin but just three had gram-positive organisms resistant to vancomycin, which is reassuring given our choice of intra-focal antibiotics. Our empirical antibiotic regimen of a glycopeptide in combination with a β-lactam antibiotic is widely accepted with its low resistance rates even in the presence of high bacterial loads, although guidelines are based on limited evidence [6]. We use teicoplanin over the more commonly used vancomycin, due to its lower risk of adverse effects including nephrotoxicity, simpler dosing regimen and less frequent requirement for drug levels. When cultures and sensitivities are reported, antibiotic therapy is tailored accordingly and oral antibiotics are considered where possible. Spellberg and Lipsky, in their 2011 review of publications investigating the use of antibiotics in chronic osteomyelitis, concluded that oral antibiotics with high bioavailability are comparable with parenteral therapy and the addition of adjunctive rifampicin may improve cure rates [28]. The use of intra-focal antibiotic delivery at the time of surgical debridement allows high-dose antibiotic delivery in difficult-to-reach tissues whilst managing dead space. Both CERAMENT^®^ G/V (BoneSupport AB, Lund, Sweden) and STIMULAN^®^ (Biocomposites Ltd.,Staffordshire, UK) have been used in FRI and chronic osteomyelitis with high rates of infection remission and union [29,30]. It is likely that good outcomes can be attributed to a combination of meticulous debridement, timely ortho-plastics management and local antibiotics. Gentamicin and vancomycin have low resistance rates with this combination of local antibiotics demonstrating a sensitivity of 94.2% in our cohort of patients.

There are important differences when considering the microbiology profile of FRI compared to PJI. Firstly, the initial damage to soft tissues overlying the surgical site, contamination of open injuries with soil microorganisms and altered vascularisation following crush injury increase the risk of a wider range of pathogens in FRI. Secondly, stability following fracture fixation is essential in both treating and preventing infection, however, it is still not clear how this impacts microbial flora [31]. Studies investigating FRI have reported polymicrobial rates of around 30% compared to the 10% in PJI [26,32]. Fungal and anaerobic infections accounted for 5.6% of organisms, which whilst lower than the 10% found by Sheehy et al., are a rare and potentially severe cause of infection [21]. In those patients where *Candida* species were isolated, no additional risk factors were identified other than the history of metalwork following open fracture. 

Our relatively high organism yield of 82.5% from culture (268/325 infection episodes where deep surgical samples were taken) may reflect our requirement for withholding antibiotics for two weeks prior to surgery and our strict standardised intra-operative sampling techniques. Onsea et al., in a multi-centre international study of 480 FRI patients, had an organism yield of 88% and similarly found the tibia and/or fibula to be the most common anatomical location for FRI [33]. 

Molecular techniques, which amplify bacterial DNA, have been shown to be a valuable diagnostic tool in PJI; however, their role in FRI is yet to be established [34]. In our laboratory, it is standard procedure for four tissue samples from any given orthopaedic surgery to be sent for 16S PCR if they are culture negative after 48 h. An identifiable bacterial species was detected from 34.9% of PCR samples (44/126). A total of 22 different bacteria were found. 45.5% (20/44) were gram-negative organisms, with *E. coli* being the most common, followed by *Pseudomonas aeruginosa*. *S. aureus* was found in three PCR positive samples (7.0%). In 36 of the 126 cases (28.6%) where PCR was done, an organism was identified which had not been cultured. In 19 of these cases, there had been no growth at all from tissue culture so 16S PCR provided the only microbiological evidence on which to base treatment. Although culture is the gold standard and has the advantage of allowing for antibiotic susceptibility testing, where samples are culture negative, 16S PCR can be invaluable in informing management plans.

Using automated laboratory techniques, cultures that contain pathogens are usually positive by day three and most by day five [35]. Since our orthopaedic samples are incubated for 48 h, our empirical antibiotic regimen of meropenem and teicoplanin is continued during this period to prevent the colonisation of freshly debrided bone surfaces. Some organisms take longer to grow requiring cultures to continue for 10–14 days [36]. Although not performed at our centre, sonication fluid culture has been shown in a systematic review to be a useful adjunct to conventional tissue culture in FRI, but not superior [37]. Currently, there is nothing in the literature to support the routine use of sonication in FRI management. 

Positive histopathological examination remains one of the confirmatory criteria in the FRI consensus definition and has recently been validated [33]. Clinicians have often resorted to extrapolating the evidence from PJI into FRI; however, Morgenstern et al. showed that the complete absence of polymorphonuclear neutrophils had a very high correlation with aseptic non-union and the presence of >5 per high-power field was always associated with FRI [38]. Our protocol includes sending a single sample for histological analysis and although this was not specifically explored in our study, we have occasionally relied on histopathological diagnosis to guide our management. 

It is well accepted that superficial wound swabs do not accurately correlate with deep tissue culture results and are consequently not considered part of the FRI gold standard of care [39]. A historical paper by Mackowiak et al. found that *S. aureus* from sinus tracts correlated with *S. aureus* in the operative specimen, but, overall, only 44% of sinus-tract cultures contained the operative pathogen [40]. We ascertained which of the patients in our FRI cohort had had superficial wound swabs in the 30 days before deep surgical samples were taken in order to see how informative superficial samples might be. Of the 96 swabs performed, 20.8% (20/96) grew the same organism(s) as later cultured from surgical tissue samples or identified on PCR. 31.3% (30/96) of samples had some similarity to surgical sample culture or PCR results, e.g., one organism was found both on swabs and deep samples but the results were not completely matched. In 45.8% (44/96) there was no relationship between swab results and deep samples. Antibiotics given in the month preceding surgery may well have affected swab culture results.

Given the lack of reliable correlation between superficial swab results and the culture of surgical samples, it would not be recommended to base treatment of FRIs on wound swab culture. However, in situations where it is not possible to obtain deep samples, and where organisms grown from wound swabs are a feasible cause of FRI, it would be reasonable to choose an empirical antibiotic regimen which covers these, as well as the organisms most implicated in such infections.

There are limitations inherent with a single-centre retrospective analysis and our microbiology profile reflects on our local population and the scope of practice of one of Europe’s busiest trauma centres. Different profiles may be encountered in other settings, and this should be taken into consideration when local guidelines are being developed. 

Despite this, we believe our study; the largest single-centre FRI case series in the literature, may guide other bone infection units in determining the most effective empirical antibiotic regimen for their patients, as well as promote further work on the topic that would not only allow other units to audit the appropriateness of their regimes but would also serve towards large data gathering to allow for better-informed choices in the management of FRI. 

## 4. Materials and Methods

All adult patients treated for FRI at our major trauma centre over a six-year period (2016–2022 inclusive) were retrospectively identified from our electronic database. All patients were treated by three consultant orthopaedic surgeons with an interest in bone infection (AI, AV, NH). We report on patient and injury characteristics, anatomical site, microbiological profile and antibiotic susceptibilities. The 2018 FRI consensus definition was used for the diagnosis of FRI [1]. 

Our bone infection service has an established pathway whereby all patients are seen pre-operatively in a combined orthoplastics clinic and are discussed in a multidisciplinary meeting with microbiologists. Our workup typically includes appropriate imaging and blood tests and anaesthetic pre-assessment when feasible. Cessation of any antibiotic therapy two weeks prior to surgical debridement was encouraged. Deep samples were taken following a strict protocol to avoid cross-contamination with at least five samples sent for microbiology and one sample for histological analysis. Tissue homogenization was achieved using Ballotini glass beads and cultured for 10 days. Identification of organisms was performed using Matrix-assisted laser desorption ionization Time-Of-Flight mass spectrometry (Maldi-TOF MS, Bruker, Bremen, Germany). Antibiotic susceptibility was tested and interpreted according to European Committee on Antimicrobial Susceptibility Testing (EUCAST) breakpoints. Our empirical antibiotic therapy was intravenous meropenem and teicoplanin following sampling and this was continued as an inpatient until sensitivities were back (usually day 3–5). Our choice of local antibiotics was a combination of gentamicin and vancomycin with the carrier either CERAMENT^®^ G/V (BoneSupport AB, Sweden), STIMULAN^®^ (Biocomposites Ltd., UK) or a combination of both. 

A positive microbiology result was defined as two or more samples with an indistinguishable organism and results were considered polymicrobial if two or more organisms were each identified in two or more samples, as per the FRI consensus diagnostic criteria [1,41]. The use of broad-range 16S ribosomal RNA (rRNA) gene polymerase chain reaction (PCR) testing was performed in cases where clinical suspicion of infection was high but routine cultures were negative. All patients were followed-up for a minimum of one year following treatment.

## Figures and Tables

**Figure 1 antibiotics-12-01358-f001:**
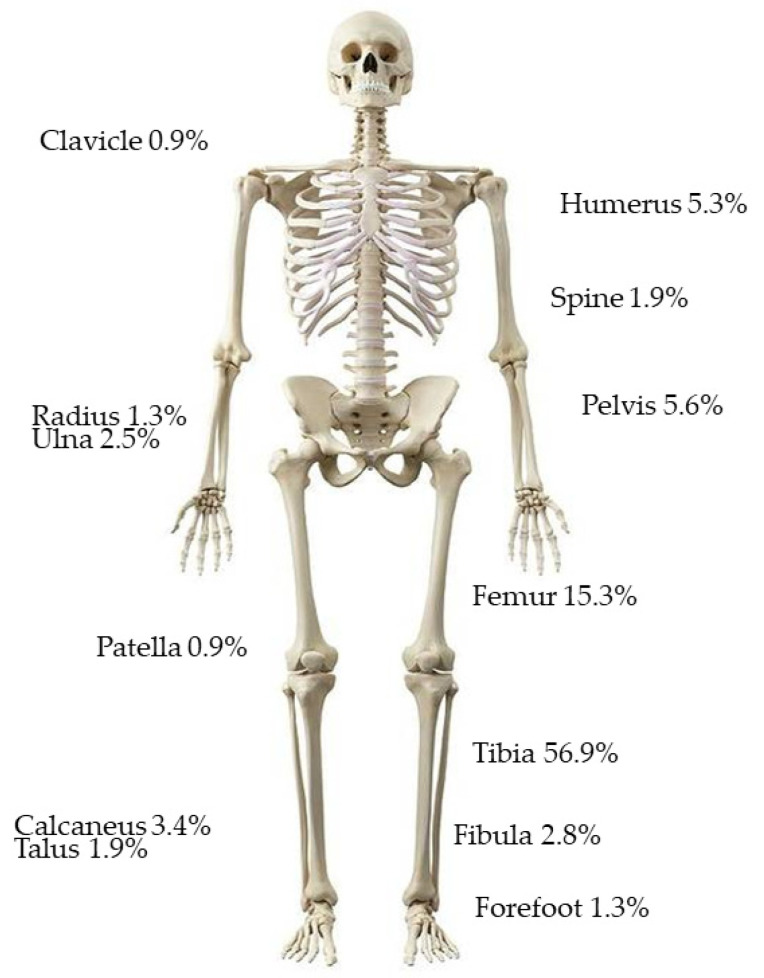
Anatomical distribution of fracture-related infection cases.

**Figure 2 antibiotics-12-01358-f002:**
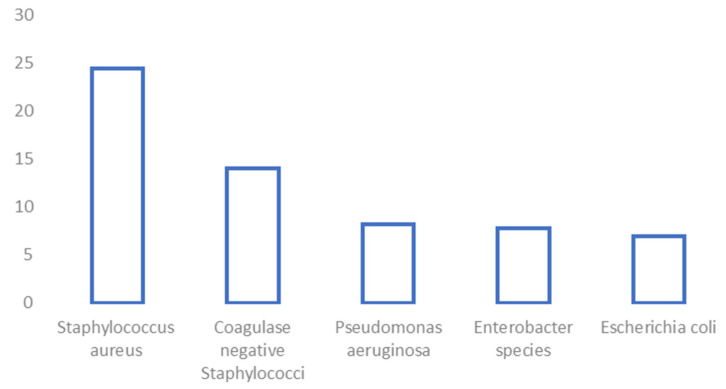
Top five isolated organisms (%).

**Table 1 antibiotics-12-01358-t001:** Fracture-related infection (FRI) patient characteristics (*N* = 294).

Demographics (%)	
Mean age at diagnosis (years)	48
Male	212 (72)
Female	82 (28)
Injury (%)	
Open	110 (37.4)
Closed	184 (62.6)
Co-morbidities (%)	
Diabetes	8 (1.6)
HIV	2 (0.6)
Drug/alcohol abuse	15 (4.7)

**Table 2 antibiotics-12-01358-t002:** Isolated organisms (*N* = 463).

Isolated Organisms			
Staphylococci	186 (40.2%)	Enterobacterales	124 (26.8%)
*Staphylococcus aureus*	113 (24.4%)	Enterobacter species	36 (7.8%)
Coagulase-negative staphylococci	65 (14.0%)	*Escherichia coli*	32 (6.9%)
		Klebsiella species	19 (4.1%)
Streptococci	21 (4.5%)	Proteus species	14 (3.0%)
Group B streptococci	8 (1.7%)	Citrobacter species	9 (1.9%)
*Strep. dysgalactiae*	8 (1.7%)	*Morganella morganii*	5 (1.1%)
Viridans streptococci	5 (1.1%)	Serratia species	4 (0.9%)
		Providencia species	3 (0.6%)
Enterococci	28 (6%)	*Salmonella typhimurium*	1 (0.2%)
*Enterococcus faecalis*	17 (3.7%)	Pantoea species	1 (0.2%)
*Enterococcus faecium*	7 (1.5%)		
Other Enterococcus species	4 (0.9%)	Non-fermenting gram-negative bacilli	42 (9.1%)
		*Pseudomonas aeruginosa*	38 (8.2%)
Other gram-positive aerobic organisms	11 (2.4%)	*Acinetobacter baumannii*	3 (0.6%)
Corynebacterium species		*Stenotrophomonas maltophilia*	1 (0.2%)
	11 (2.4%)		
Gram-positive anaerobes		Gram-negative anaerobes	7 (1.5%)
Peptostreptococcus species	18 (3.9%)	Bacteroides species	6 (1.3%)
*Finegoldia magna*	8 (1.7%)	Prevotella species	1 (0.2%)
Actinomyces species	4 (0.9%)		
Clostridium species	3 (0.6%)	Fungal	8 (1.7%)
*Cutibacterium acnes*	2 (0.4%)	*Candida albicans*	4 (0.9%)
	1 (0.2%)	*Candida dublinensis*	1 (0.2%)
		*Candida glabrata*	1 (0.2%)
		*Aspergillus fumigatus*	1 (0.2%)
		*Rhizopus arrhizus*	1 (0.5)

**Table 3 antibiotics-12-01358-t003:** Culture results (%).

Total Infection Episodes = 325	
No significant growth (on cultures or PCR)	57 (17.5)
Single-organism growth	157 (48.3)
Polymicrobial	111 (34.2)
Total potentially pathogenic organisms isolated = 463	
Gram +ve	265 (57.2)
Gram −ve	184 (39.7)
Gram-variable	1 (0.2)
Mixed anaerobes	5 (1.1)
Fungal	8 (1.7)
Polymerase Chain Reaction (PCR) samples = 126	
Organism identified	44 (34.9)
Organism identified on PCR alone (culture negative)	19 (15.1)
Superficial swabs = 96	
Same organism as deep culture/PCR	20 (20.8)
No similarity to deep culture/PCR	44 (45.8)

**Table 4 antibiotics-12-01358-t004:** Infection episodes involving resistant organisms.

Methicillin-resistant *Staphylococcus aureus*	8 (2.4) 113 7.1% of *S. aureus* isolated
Vancomycin-resistant Enterococci	3 (0.9) 10.7% of 28 Enterococcus species isolated
Extended spectrum beta-lactamases	14 (4.2)
ampC	29 (8.8)
Carbapenemase-producing organisms	3 (0.9)
Other multi-drug resistant organisms	1 (0.3)

**Table 5 antibiotics-12-01358-t005:** Infection episodes involving organisms resistant to empiric treatment.

Meropenem-resistant gram-negatives	5 (1.5%)
Gentamicin-resistant gram-negatives	16 (4.8%)
Teicoplanin-resistant gram-positives	1 (0.3%)
Vancomycin-resistant gram-positives	3 (0.9%)

## Data Availability

No new data were created or analyzed in this study but our raw anonymized data is available on request.

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
