# Peer review of "Microbiological Profile of Fracture Related Infection at a UK Major Trauma Centre"

_antibiotics, 2023, doi:10.3390/antibiotics12091358_

Round 1

Reviewer 1 Report

The authors performed a study with the aim to characterize the microbiological profile of fracture-related infection in a large patient cohort.

Overall, it is an interesting and well performed study. This said, there are some issues that need to be addressed. 

General remarks:

First of all, there is a recent paper by Depypere et al. that is highly relevant to this study as it has a similar aim and study set up: PMID 35873162. Please refer to this study in your manuscript and use it in the discussion.

The FRI consensus definition was updated in 2019. Please also refer to this reference: PMID 31855973.

The majority of fractures was of the tibia/fibula. Do you see a different microbiological distribution compared to the other sites? Were most fractures in this anatomical region open?

In comparison to the study with PMID 35873162, the rate of culture-negatives (17.5%) is quite high in the presented study. What was the clinical presentation in these patients?

31 patients had more than one infection episode at least 30 days apart. How was this defined? Was this an infection at the same anatomical site?

As no cultures were taken in 5 patients, I would suggest to exclude these patients from the analysis, and only report on the remaining 325 episodes.

The time-based classification that was used in this study does not seem relevant. Most studies refer to the Willenegger and Roth classification, subdividing FRI into early, delayed and late onset infections, occurring within 2, 3-10 or >10 weeks. Others describe a cut-off of 6 weeks to differentiate between acute and chronic infections. However, classifying FRIs based on time-related cut-offs remains arbitrary. On which basis was the time cut-off of 1 year chosen?

Please write numbers below ten in full (‘1’ as ‘one’,…)

Please revise the lay-out of all tables as they are difficult to interpret. It would be helpful to list the variables below each other instead of next to each other.

Staphylococcus aureus should be abbreviated as S. aureus. Please revise this throughout the manuscript.

Swab cultures were used in this study for some patients as an ‘extra’ or ‘adjunct’ to deep tissue cultures. The authors acknowledge the lack of reliable correlation between superficial swab results and deep tissue culture results. Therefore it seems incorrect to suggest to base an antibiotic regimen on swab culture results. Please adjust this in the text (lines 243-248).

Minor remarks:

Abstract:

Please be consistent with the terminology throughout the manuscript: ‘microbial profile vs. microbiological profile’.

Introduction:

Line 44: ‘optimum’ revise to ‘optimal’.

Lines 45-46: ‘This is especially relevant in those cases treated empirically or when culture results don’t yield a pathogen.’ Please revise to: ‘… or when culture results are negative.’

Line 49: Please refer to the recent multicenter study as mentioned above: PMID 35873162.

Lines 56-59: This sentence seems to be a bit out of place. As the aim of the study is to describe the microbiological profile, it would perhaps be more relevant to describe the micro-organisms that were reported in this case series, rather than the infection rate. It is well-established that the infection rate after severe open fractures can increase up to 30%.

Line 60: Please delete the extra space between ‘, the authors’ and ‘reported that most FRI cases…’.

Line 74: Please revise to ‘in the PJI group (11.6 vs 2.3%).’

Lines 98-99: Please use the abbreviation ‘FRI’ or write it in full throughout the manuscript.

Results:

Lines 103-108: See general remarks.

Table 1: Please revise this table, list the variables mean age, male, female below each other. Similar for the other variables under Injury, Co-morbidities and Time to diagnosis.

I would suggest to include the GA grade for the open fractures and perhaps link this to the microbiological profile.

Table 2: Same remark for all tables, please update the lay-out. Furthermore, the percentages in the second column are not aligning properly with the first column. Perhaps this data could be presented graphically in pie charts: e.g. a pie chart showing the distribution of the bacterial families (Staphylococci, Streptococci, Enterococci, Other gram positive aerobic organisms, Gram positive anaerobes, Enterobacterales, Non fermenters, Gram negative anaerobes, Fungi). This could potentially replace or be a part of Figure 2.

Table 3: Same remark for all tables, please update the lay-out.

Table 4: Same remark for all tables, please update the lay-out.

Discussion:

Lines 152-153: In this sentence you refer to Depypere et al for the incidence of GNB, however you did not include a reference. In the study with PMID 35873162, this author reports an incidence of 25.3%. Please revise this accordingly.

Line 174: Please add an extra space between after the full stop, prior to ‘The use of intra-focal…’. Delete the extra space between ‘intra-‘ and ‘focal’ and between ‘delivery’ and ‘at’.

Materials and Methods:

Please add detail on the applied microbiological culture methods. 

I have no specific comments regarding the quality of English language. I've made some suggestions regarding typos.

Reviewer 2 Report

The work is well written, the methodology is adequate, and their experience is relevant as large multicenter studies focused on the microbiology of FRI are lacking so it may help will help to FRI better manage. Therefore, I consider that the article deserves to be published with only minor changes.

 However, I would suggest to the authors consider the following minor revisions:  

1- Table 2 (page 4) needs to be reviewed and corrected since the lines have been moved and the numerical values do not correspond to the variables with which they are associated

2- I do not see Figure 2 (page 5) as necessary since the information it provides is described in the text. Perhaps it could be eliminated from the work.

3- Again table 3 (page 5) needs to be reviewed since it seems that the registered variables have moved

4- I also advise separating tables 3 and 4 (they appear linked to each other in the draft)

The discussion, although extensive, is well-argued and interesting, so I do not recommend changes.

Reviewer 3 Report

Very well done analysis. I have no comments

Round 2

Reviewer 1 Report

The authors have considered all previous comments and have made adaptations to the text, which have improved the quality of the manuscript. I would therefore recommend the current manuscript for publication.